# Graphene Nanoribbon Field Effect Transistor Simulations for the Detection of Sugar Molecules: Semi-Empirical Modeling

**DOI:** 10.3390/s23063010

**Published:** 2023-03-10

**Authors:** Asma Wasfi, Ahmed Al Hamarna, Omar Mohammed Hasani Al Shehhi, Hazza Fahad Muhsen Al Ameri, Falah Awwad

**Affiliations:** 1Electrical and Communication Engineering Department, College of Engineering, United Arab Emirates University, Al Ain P.O. Box 15551, United Arab Emirates; 2Chemical and Petroleum Engineering Department, College of Engineering, United Arab Emirates University, Al Ain P.O. Box 15551, United Arab Emirates; 3Mechanical and Aerospace Engineering Department, College of Engineering, United Arab Emirates University, Al Ain P.O. Box 15551, United Arab Emirates

**Keywords:** GNR-FET, glucose, xylose, fructose, sensor, graphene, semi-empirical calculations, non-equilibrium green’s function (NEGF)

## Abstract

Graphene has remarkable characteristics that make it a potential candidate for optoelectronics and electronics applications. Graphene is a sensitive material that reacts to any physical variation in its environment. Due to its extremely low intrinsic electrical noise, graphene can detect even a single molecule in its proximity. This feature makes graphene a potential candidate for identifying a wide range of organic and inorganic compounds. Graphene and its derivatives are considered one of the best materials to detect sugar molecules due to their electronic properties. Graphene has low intrinsic noise, making it an ideal membrane for detecting low concentrations of sugar molecules. In this work, a graphene nanoribbon field effect transistor (GNR-FET) is designed and utilized to identify sugar molecules such as fructose, xylose, and glucose. The variation in the current of the GNR-FET in the presence of each of the sugar molecules is utilized as the detection signal. The designed GNR-FET shows a clear change in the device density of states, transmission spectrum, and current in the presence of each of the sugar molecules. The simulated sensor is made of a pair of metallic zigzag graphene nanoribbons (ZGNR) joint via a channel of armchair graphene nanoribbon (AGNR) and a gate. The Quantumwise Atomistix Toolkit (ATK) is used to design and conduct the nanoscale simulations of the GNR-FET. Semi-empirical modeling, along with non-equilibrium Green’s functional theory (SE + NEGF), is used to develop and study the designed sensor. This article suggests that the designed GNR transistor has the potential to identify each of the sugar molecules in real time with high accuracy.

## 1. Introduction

Sugar molecule detection is a critical field of research for various applications such as food quality evaluation and blood sugar detection. Sugars such as glucose, fructose, and xylose are necessary organic substances for both plants and people as they play a number of crucial roles in their growth and life. Glucose and fructose are vital elements in the human diet. They can be introduced as additives, or they can be found naturally in a variety of foods. The detection of fructose and glucose is highly critical for food quality evaluation, such as for fruit juices, fruit, and honey. Reliable and quick detection of sugar molecules is needed in various situations, e.g., fruit maturation in the field, food production, and food storage [1].

Fructose is a significant dietary source of carbohydrates that has a role in the development of diabetes problems and may contribute to the epidemic of obesity [2]. The detection of fructose can benefit the food sector and help in the diagnosis of diabetic symptoms [2]. Xylose is a type of sugar found in urine or blood samples. Xylose levels are checked to determine if there is an issue with people’s ability to absorb nutrients. Glucose identification can be used in several applications, including chemical and biological analyses [3,4,5,6], industrial applications [7,8], food industries [9], environmental [10,11,12], and clinical diagnostic applications [13,14].

Diabetes is a chronic metabolic condition that impacts blood sugar levels and is one of the most widespread and deadly diseases in the world, impacting millions of individuals. For efficient diabetic treatment, blood glucose levels must be easily and reliably monitored [15]. Continuous blood glucose monitoring can lower the risk of problems brought on by hypoglycemia or hyperglycemia, allowing for the proper treatment of diabetes [15].

Two-dimensional graphene, including functionalized graphene nanomaterial, is thought to be the ideal option for glucose sensors because the performance of biosensors depends on their surface to promote charge transfer. Platinum-functionalized graphene was utilized to detect glucose with a 0.6 M detection limit in [16]. Additionally, gold nanoparticles were investigated to find 0.3 M glucose concentration in [17]. For the detection of glucose, a variety of nanomaterial biosensors, including carbon nanotubes and graphene, have been utilized [18,19]. Designing electrochemical reaction sensors based on nonenzymatic glucometers has been researched using a variety of technologies, including carbon-based materials such as reduced graphene oxide (rGO), graphene, metal nanoparticles [20,21], and carbon nanotubes (CNT) [22,23].

Graphene-based biomolecule sensors identify the target biomolecule via different mechanisms. One of the most common detection techniques is detecting the electrical current variation in field effect transistors (FET) at a specific gate voltage to detect the target molecule. Graphene-based FET has been used to identify and detect a wide range of biomolecules, gases, toxic compounds, and chemicals with a higher sensitivity than solid-state sensors. The graphene-based FET sensitivity ranges from parts per billion (ppb) to parts per million (ppm) [24,25,26,27,28]. The gate terminal controls the current flow through graphene-based FET biosensors.

The biomolecule adsorption affects the charge carrier’s concentration through the graphene channel, and consequently, the sensor current varies at a specific gate voltage. Some of the target biomolecules increase the FET current by acting as donors, while other target molecules reduce the sensor current and act as acceptors. This electrical current change is used as the detection signal of the target biomolecule [29,30,31]. Moreover, the conductance fluctuations can also be utilized as the detection signal [32].

Various material processing methods are utilized to fabricate graphene materials with exceptional electronic characteristics, such as carbon nanoribbons, carbon nanotubes, and carbon nanosheets. Graphene nanoribbons (GNRs) can be fabricated by unzipping carbon nanotubes via various techniques [33]. The electrical characteristics of Graphene nanoribbon-based FET are interesting and change with the direction and the width of the nanoribbons [34,35].

This work is innovative, as it utilizes a GNR-FET for the first time as a sensor to detect and differentiate between sugar molecules—glucose, fructose, and xylose. As far as we are aware, this is the first research that employs a FET composed of an AGNR channel and a set of metallic ZGNR electrodes to recognize these sugar molecules. 

This study has several potential implications and contributions to the research community. These include: (1) Advancing the field of graphene-based sensors: The study highlights the potential of graphene-based sensors as a reliable and sensitive tool for detecting and differentiating between different types of sugar molecules. This could inspire further research into the use of graphene and related materials for sensing applications and potentially lead to the development of new and improved sensor technologies. (2) Improving our understanding of charge transfer mechanisms: The study sheds light on the charge transfer mechanisms that occur when sugar molecules interact with graphene, specifically how the molecules act as donors and increase the concentration of charge carriers. This could help deepen our understanding of fundamental processes related to charge transfer and doping in graphene-based materials.

In this work, a GNR-based FET device is simulated via an Quantumwise Atomistix Toolkit (ATK). The designed device is utilized to identify the presence of fructose, xylose, and glucose molecules. The current variation through the GNR-FET in the presence of each sugar molecule is utilized as the detection mechanism. A semi-empirical model, along with non-equilibrium Green’s function (SE + NEGF), is used to examine the designed sensor. Several electronic properties are studied, such as density of state, transmission spectrum, and electrical current. The GNR monolayer’s properties hold promise for sugar molecule sensing devices.

## 2. Materials and Methods

The GNR-FET simulations for the identification of sugar molecules such as fructose, glucose, and xylose were conducted via Quantumwise Atomistix Toolkit (ATK), and its graphical user interface Virtual Nano Lab (VNL). ATK-VNL provides atomic-scale modeling and calculations for nanoscale systems. The software provides multiple built-in calculators to generate and calculate electron transport characteristics of quantum systems. A semi-empirical (SE) calculator was utilized to generate the simulation for the proposed GNR-FET. ATK-SE mathematical formalism and workflow was given in ATK-SE [36]. A high performance computing environment (HPC) was utilized to run the simulation and generate the results. Seven computing nodes were used via HPC, where each node has 36 processing units. Transmission Spectra, electrical currents, and device density of states were generated in the presence of glucose, fructose, and xylose.

### 2.1. Graphene Nanoribbons (Armchair and Zigzag)

The GNR-FET consists of armchair graphene nanoribbons (AGNR) and zigzag graphene nanoribbons (ZGNR). The nanoribbon type is determined by the structure edges’ termination pattern, either a zigzag edge or an armchair edge [37,38]. Figure 1a depicts a zigzag graphene nanoribbon because the red termination edge shows a zigzag pattern, while Figure 1b depicts an armchair graphene nanoribbon because the red termination edge shows an armchair pattern. The bandgap of the GNR differs with the variation in the number of carbon atoms [38]. The increment of the number of carbon atoms in the AGNR and ZGNR structures leads to a decrement in the bandgap [38]. GNRs enable the design of a perfect junction at atomic scale levels. It is challenging to avoid contact resistance levels between the molecular device and the metal electrodes due to the limited contact surfaces. The solution to this problem is to use metallic GNRs that can be connected to the circuits directly [35,39].

### 2.2. Sensor Setup and Configuration

The GNR-FET consists of armchair graphene nanoribbons (AGNR) and zigzag graphene nanoribbons (ZGNR). The GNR-FET is simulated by utilizing ZGNR and AGNR in ATK-VNL. The ATK builder tool is utilized to design and simulate the sensor. The central area of the FET is made of AGNR by the Nanoribbon Plugin Tool with a width of four atoms. The electrodes are made of ZGNR by the Nanoribbon Plugin Tool with a width of six atoms. The two ZGNR electrodes are joined via the AGNR to form the z-shaped structure. The left and right electrodes are made of ZGNRs. A gate made of two layers (dielectric layer and metallic layer) is added underneath the central region of the z-shaped structure to form a FET sensor, as shown in Figure 2. The dielectric material permittivity is set to 3.9 ε. The electrode length is 7.4 Å while the channel length is 19.2 Å. Each of the sugar molecules displayed in Figure 3 is exposed to the GNR-FET device. The change in device density of states, current, and transmission spectra are generated in the presence of glucose, xylose, and fructose. Figure 4 shows the GNR-FET device structure with xylose.

### 2.3. Computational Method

There are 112 atoms in the system., Semi-empirical electron transport calculations were performed to determine the electronic signature of each sugar molecule. The performance of the sensor was examined using both the extended Hückel method and the non-equilibrium Green’s Function formalism. The finite bias voltage, which ranged from 0 to 2 V, was fixed among the left and right electrodes and the gate voltage was fixed at −1 V.

5 × 5 × 100 k-points were used as the sampling k-points for the Brillouin Zone integration. To represent the charge density, a mesh cut from 10 Hartree was fixed. The electrodes’ electrostatic potential boundary conditions in the C direction were chosen to be a Dirichlet condition, while the electrodes’ electrostatic potential boundary conditions in the A and B directions were chosen to be Neumann conditions. The electrostatic voltage was fixed using the Neumann boundary condition, resulting in a derivative with zero [40].

A semi-empirical method was utilized to generate the electronic transport properties of the GNR-FET sensor with each of the sugar molecules. The electronic transport properties were examined using semi-empirical approaches along with non-equilibrium Green’s function (NEGF). The semi-empirical methodology used involved the extended Hückel (EH) approach, combined with the self-consistent (SC) Hartree potential. The NEGF+SC-EH was used to demonstrate that each of the four sugar molecules placed on the device channel produces a distinct electrical signature.

The following formula was used to obtain the zero bias transmission spectrum between the source and drain [41,42]:(1)$$T\left(E\right)=Tr\left\{\Gamma_{D}\left(E\right)G\left(E\right)\Gamma_{S}\left(E\right)G^{ϯ}\left(E\right)\right\}\;$$
where E denotes the energy,  Tr denotes the trace, $\Gamma_{D,\;S}\left(E\right)=i\left[\sum_{L,\;R}\left(E\right)-\sum_{S,\;D}^{ϯ}\left(E\right)\right]$ specifies the broadening level because of the coupling to the electrodes, and $\sum_{L,\;R}\left(E\right),\;\;\sum_{S,\;D}^{ϯ}\left(E\right)\;$are the self-energies presented by the electrodes.

The NEGF method, which is integrated into ATK-VNL, was utilized to analyze the electron transmission spectrum as a function of bias using [41,43]:(2)$$T\left(E,V_{b}\right)=Tr\left\{\Gamma_{D}\left(E,V_{D}\right)G\left(E\right)\Gamma_{S}\left(E,V_{S}\right)G^{ϯ}\left(E\right)\right\}\;$$
where *G* and $G^{ϯ}$ are associated with advanced Green’s function of the main scattering region, and $V_{b}=V_{s}-V_{d}$ where $V_{b}$ is the bias voltage across the source ($V_{s}$) and drain ($V_{d}$). *S*, *D*, *L*, and *R* refer to the source, drain, left, and right, respectively. 

The integration of T(E, V) over the energy window was determined using the difference between the Fermi functions $f_{S,\;D}\left(E\right)={\left\{1+\exp\left[\left(E-E_{F}-eV_{S,D}\right)/k_{B}T\right]\right\}}^{-1}$ which gives the total current [41]:(3)$$I=\frac{2e}{h}\overset{\infty }{\underset{-\infty }{\int }}dE\;T\left(E,V\right)\left[f_{S}\left(E\right)-f_{D}\left(E\right)\right]\;$$

## 3. Results and Discussion

The electrical transport parameters for the GNR-FET were generated to enable the practical study of the designed GNR-FET sensor to precisely detect each of the sugar molecules.

### 3.1. Device Density of State (DDOS)

The presence of each of the sugar molecules has caused a clear and noticeable change in the FET Device DOS. Figure 5 compares the DDOS for the GNR-FET with and without each sugar molecule. Figure 5 demonstrates that the bare GNR-FET has more energy states than the GNR-FET when a sugar molecule is present. The GNR-FET energy spikes are at −1.6, 0, 1.75, and 1.9 eV. The presence of fructose has initiated a new energy spike at 0.65, as shown in Figure 5a, while the presence of xylose resulted in low energy spikes at −0.9, 0.8, and 1.6, as displayed in Figure 5b. Similarly, the GNR-FET exhibits a considerable shift in DDOS when exposed to the glucose molecule, as seen in Figure 5c.

The partial DOS in Figure 6 provides detailed information on how each sugar molecule affects the DDOS. Upon adding a target molecule to the device, a unique peak in the DDOS is increased due to the presence of glucose (Figure 6a), fructose (Figure 6b), or xylose (Figure 6c). The addition of these sugar molecules introduces new electronic states within the energy range of the peak, potentially indicating an interaction between the sugar molecule and the armchair GNR channel. This interaction could modify the electronic structure of the channel, resulting in changes to the DDOS. The changes to the DDOS could be due to the sugar molecule accepting or donating electrons from the channel material, or through the formation of chemical bonds between the target molecule and the AGNR channel.

### 3.2. Transmission Spectra

Figure 7 shows the transmission spectra T(E) for the GNR FET at several biases, including (a) V = 0 V, (b) V = 1 V, and (c) V = 2 V, without and with each of the sugar molecules (glucose, fructose, and xylose). The figure demonstrates how changing the applied voltage and adding additional sugar molecules affect a transmission spectrum. The energy window within the band gap of the semiconducting GNR channel causes the transmission spectrum to have a low value in the [0.1, 0.4] eV energy range.

The transmission spectrum of GNR-FET is not expected to be symmetrical, because of the presence of an electric field applied by the gate voltage (−1 V) and the geometry of the device (z-shaped). 

The graphene nanoribbon (GNR) acts as the channel through which electrons flow between the source and drain electrodes. The gate voltage is applied to control the electron density in the channel and thus modulate the conductance of the device. When the gate voltage is applied, it creates an electric field perpendicular to the graphene plane, which can affect the transport properties of the GNR. 

Additionally, the z-shaped geometry of the device can also contribute to the asymmetry of the transmission spectrum. The z-shaped GNR-FET has two different regions where the GNR is in contact with the electrodes, which can have different electronic properties due to the different edge configurations. This can lead to an asymmetric transmission spectrum, as the electron transport through each of these regions can be different.

The combination of the gate voltage and the z-shaped geometry of the GNR-FET can lead to an asymmetric transmission spectrum, which is expected and a characteristic of this type of device. These results are consistent with previous studies [44,45].

The connection between the transmission spectrum and the DOS can be understood in terms of the Landauer–Büttiker formula, which relates the conductance of a quantum system to the transmission probability and the DOS of the electrodes. In this formula, the transmission probability is integrated over all energies, weighted by the DOS of the electrodes. This relationship implies that the transmission spectrum and the DOS are closely related and that changes in the DOS can affect the transmission spectrum.

Therefore, the transmission spectrum and the density of states are both important quantities that can be used to study the electronic properties of materials and devices and are related to each other through the Landauer–Büttiker formula.

### 3.3. Current-Voltage

In order to calculate the current from the transmission spectrum in QuantumWise, the software uses the Landauer–Büttiker formalism. The Landauer–Büttiker formalism is a widely used approach for calculating the conductance and current through a quantum device. The formalism is based on the transmission probability of electrons through the device, which is calculated using a quantum mechanical description of the device.

The transmission probability is calculated by solving the Schrödinger equation for the electronic wave function in the device. This involves constructing a Hamiltonian matrix that describes the interactions between the electrons and the atoms in the device and then solving the eigenvalue problem to obtain the energy levels and wave functions of the system.

Once the transmission probability is obtained, the Landauer–Büttiker formalism can be used to calculate the current flowing through the device, as displayed in Equation (3). 

In practice, the integration over energy is typically performed numerically using a discrete set of energy values that span the relevant energy range. The chemical potentials of the electrodes can be set by specifying the applied bias voltage or by constraining the total number of electrons in the system [46].

Drain-to-source current (I_ds_) and voltage (V_ds_) curves of the designed GNR-FET were generated for the fixed gate voltage of −1.0 V. Figure 8 shows an increase in I_ds_ due to the presence of each of the sugar molecules. When target sugar molecules are present, the IV curves of FET devices are found to significantly differ from those of the device when target sugar molecules are not present. This variation in IV characteristics can be exploited as a detection signal. The target sugar molecules serve as charge carriers, acceptors, or donors for graphene. The concentration of the graphene charge carriers increases when the target molecule act as donors. As a result, under the same biased voltage and gate voltage, the current flowing via the graphene-based sensor increases in the presence of sugar molecules.

The sugar molecules act as charge carriers, specifically donors, because the current flowing through the GNR-FET increases in the presence of these molecules. When the sugar molecules are introduced to the transistor channel, they provide extra electrons to the graphene, which in turn increases the concentration of charge carriers.

In terms of the Fermi level, the introduction of donor molecules leads to a shift in the Fermi level towards the conduction band of the graphene, resulting in an increase in the concentration of electrons available for conduction. The sugar molecules act as n-type dopants. This is because they introduce extra electrons into the graphene, which are negatively charged and contribute to the overall electron concentration.

The change in the current reading due to the addition of the sugar molecules indicates a successful detection. The adsorbed target molecule interacts with the GNR-FET channel and changes its conductivity by changing the carriers’ concentration. GNR is a semiconducting material which has a nonlinear resistance resulting in a nonlinear IV curve, as shown in Figure 8. The addition of the sugar molecules causes a change in the GNR-FET sensor’s current, which proves the successful detection of the sugar molecules.

The GNR-electrical FET’s current varies differently for each sugar molecule. Every sugar molecule differs in size, electrical state, and how it interacts with the GNR-FET channel. The best results are generated when the gate potential is fixed at −1 V, and the bias voltage among the left and right electrodes is fixed at 2 V. Figure 9 demonstrates that the best sensitivity is achieved by setting the bias voltage to 2 V. This research establishes the feasibility of using the created GNR-FET to identify various sugar molecule kinds.

The sensor displays the highest sensitivity at a fixed bias of 2 V. The sensor’s response (variation in current) is shown in Figure 9, where the highest variation in the electrical signal is due to the presence of the glucose molecule. These findings show that the sensor is highly selective for glucose and generates a distinct electrical signal for each sugar molecule.

The higher sensitivity of the GNR-FET sensor to glucose compared to fructose and xylose could be attributed to several factors, such as the differences in the chemical structure and electronic properties of the sugar molecules. Glucose has a six-membered ring structure, while fructose and xylose have five-membered ring structures. The differences in the ring size could result in variations in the electronic properties of the molecules, which could affect their interaction with the GNR-FET sensor.

Moreover, the position of the hydroxyl (-OH) groups attached to the ring structure of the sugar molecules could also play a role in the differences in sensitivity. Glucose has a hydroxyl group in the axial position, while fructose and xylose have hydroxyl groups in the equatorial position. This difference in the orientation of the hydroxyl groups could lead to variations in the interaction between the sugar molecules and the GNR-FET sensor.

In addition, the differences in the number and position of the -OH groups on the sugar molecules could also contribute to the variation in sensitivity. Glucose has five -OH groups, while fructose has six -OH groups, and xylose has four -OH groups. The variation in the number of -OH groups could affect the electrostatic interaction between the sugar molecules and the GNR-FET sensor, which could lead to differences in sensitivity.

Figure 10 shows that an increment in the GNR-FET current occurrs when two glucose molecules are placed on the channel. The graph shows that when two glucose molecules are introduced onto the channel, there is a noticeable increase in the electrical current passing through the transistor. This result is consistent with the operation of the GNR-FET sensor, which typically works by detecting changes in the charge distribution near the transistor’s surface caused by the binding of target molecules (in this case, glucose) to a sensing element on the transistor’s surface. The binding of the glucose molecules to the sensing element causes a change in the local charge distribution, which in turn affects the electrical properties of the transistor and leads to a measurable change in the electrical current passing through it.

Overall, the result shown in the figure suggests that the GNR-FET biosensor is capable of detecting the presence of glucose molecules in a sample, and that the sensitivity of the biosensor can be tuned by adjusting the number of sensing elements on the transistor’s surface.

If the Dirac point were to be measured for the GNR-FET, it could potentially be used as a sensing method for detecting changes in the electronic properties of graphene-based FET. The shift in the Dirac point could indicate changes in the doping level or charge carrier concentration, which could be correlated with the presence or absence of specific molecules. Nonetheless, further research would be required to explore the feasibility of using the Dirac point shift as a sensing method for detecting changes in the electronic properties of graphene-based sensors.

Compared to the method used in this study (detecting current variation), using the Dirac point to detect each sugar molecule would be more time-consuming and computationally expensive. It may be more practical to consider other sensing methods, such as measuring changes in the GNR-FET current, the density of states, or the transmission spectrum. These methods may be more straightforward and easier to implement experimentally.

In a previous study, a high I_ON_/I_OFF_ ratio of approximately 2000 was reported for a Z-shaped graphene nanoribbon FET by Yan et al. [39]. The perfect atomic interface between the metal and semiconductor graphene nanoribbons, resulting in minimum contact resistance, was attributed to this high ratio. 

In this study, the focus was on characterizing the structural and electronic properties of the GNR FET to identify a unique electronic signature for each of the sugar molecules. Due to time and resource constraints, the transconductance was not calculated within the scope of this study.

A distinct electronic signature that can be identified using the designed GNR-FET was found for each of the sugar molecules in this research. By analyzing the device’s transmission spectra, current, and density of states, unique features for each of the sugar molecules that can potentially be used for their detection and discrimination were identified.

Insights into the electronic properties of the designed GNR-FET and its potential for sensing applications were provided by this study. 

The computational method used in this study provides valuable information about the performance of the designed sensor. However, it is important to note that this method alone cannot determine the limit of detection in real-time applications. In order to identify the limit of detection, it is necessary to conduct experiments by measuring the response of the fabricated sensor to various concentrations of the target analyte.

In order to overcome the limitations of this technology, it is recommended to combine the computational method with experimental data. This approach allows for a comparison of the results obtained from both methods, which can help identify potential sources of error and uncertainty. Moreover, the use of computational methods can guide future studies aimed at enhancing the sensor’s performance, which can then be tested experimentally.

Overall, a combination of computational and experimental approaches can provide a more comprehensive understanding of the sensor’s performance and its potential for real-time applications.

## 4. Conclusions

Real-time detection of sugar molecules is essential to monitor and prevent diabetes as well as to assess food quality. In this study, a graphene nanoribbon-based field effect transistor (GNR) was built and examined to detect sugar compounds, including xylose, fructose, and glucose. The sensor was investigated using non-equilibrium Green’s function and the extended Hückel approach. Investigations were conducted on several electronic properties, including electrical current, transmission spectrum, and density of state. The proposed sensor is made up of two ZGNRs joined together by an AGNR channel and a gate. The target molecules’ presence caused a change in the GNR-FET’s electronic properties. The ability of the GNR-FET sensor to identify the sugar molecules is supported by the quantifiable differences in the electrical properties caused by each of the sugar molecules. The focus of the study was to investigate the electronic signatures of different sugar molecules in the GNR-FET and demonstrate their differentiation based on the electrical response of the sensor. However, it is agreed that the accuracy of the sensor is a crucial parameter in any practical sensing application. Further experimental studies can be done to evaluate the accuracy and sensitivity of the GNR-FET sensor in detecting sugar molecules. This will involve measuring the response of the sensor to various concentrations of sugar molecules to determine the limit of detection and the dynamic range of the sensor. It is believed that these future studies will provide valuable insights into the practical application of the sensor for sugar molecule sensing.

## Figures and Tables

**Figure 1 sensors-23-03010-f001:**
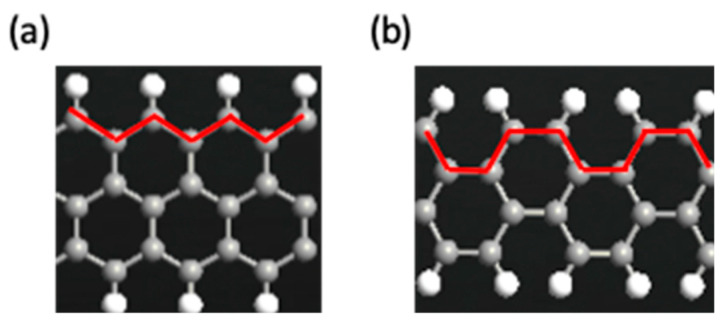
Schematic diagram of (**a**) zigzag graphene nanoribbon (**b**) armchair graphene nanoribbon.

**Figure 2 sensors-23-03010-f002:**
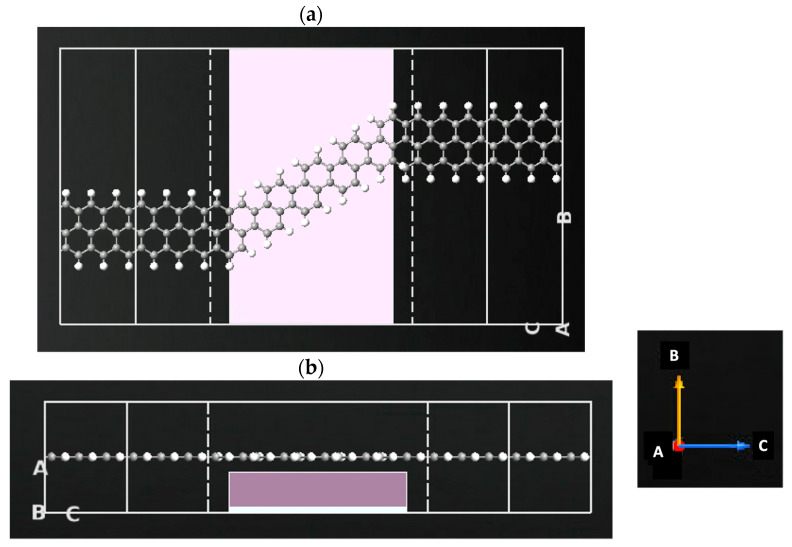
GNR-FET sensor designed by ATK-VNL. (**a**) Schematic representation of the GNR-FET sensor. (**b**) Cross-sectional view of the GNR-FET sensor. The designed sensor is made of a pair of metallic ZGNR (source and drain), an AGNR channel and a gate underneath the channel. Color code: hydrogen-white, carbon-gray.

**Figure 3 sensors-23-03010-f003:**
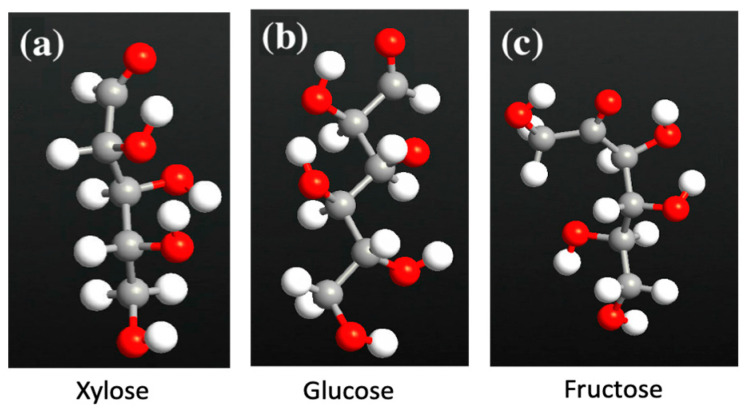
Sugar molecules’ atomic structure: (**a**) xylose, (**b**) glucose, and (**c**) fructose. Color code: oxygen-red, hydrogen-white, carbon-gray.

**Figure 4 sensors-23-03010-f004:**
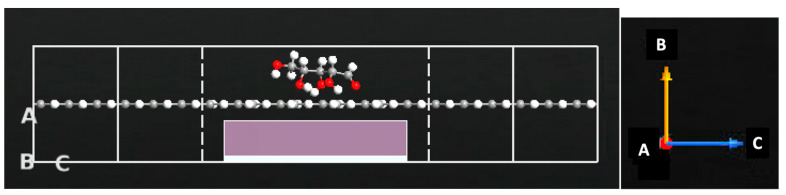
Cross-sectional view of the GNR-FET device with xylose.

**Figure 5 sensors-23-03010-f005:**
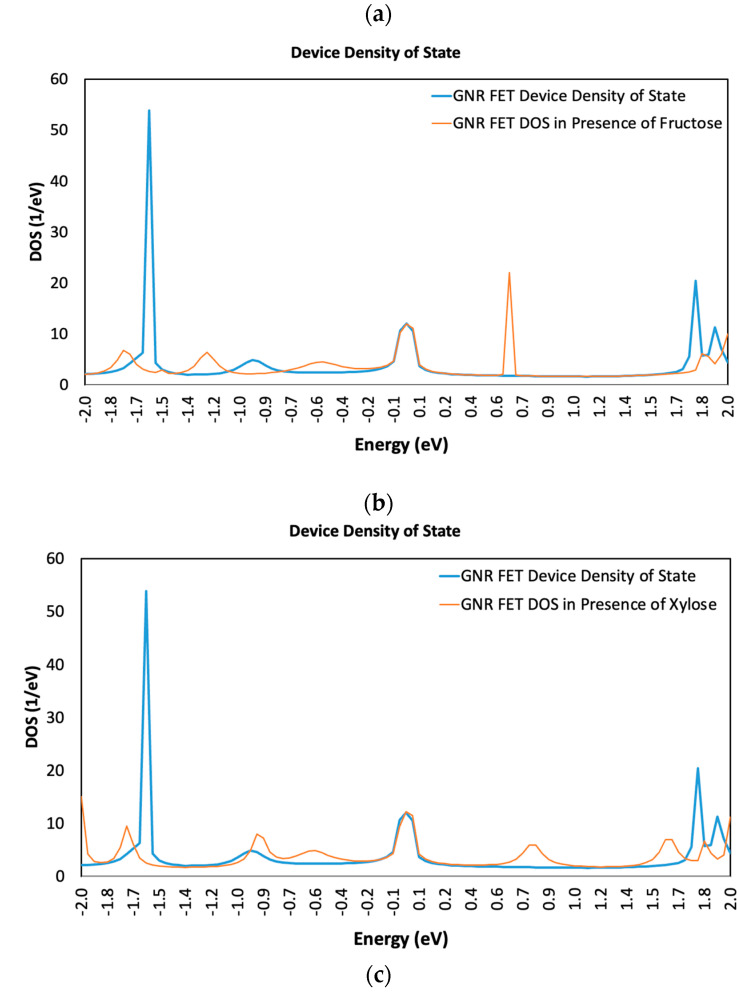
Variation in device density of states (DDOS) of simulated GNR-FET in presence of (**a**) Fructose molecule; (**b**) Xylose molecule; and (**c**) Glucose molecule.

**Figure 6 sensors-23-03010-f006:**
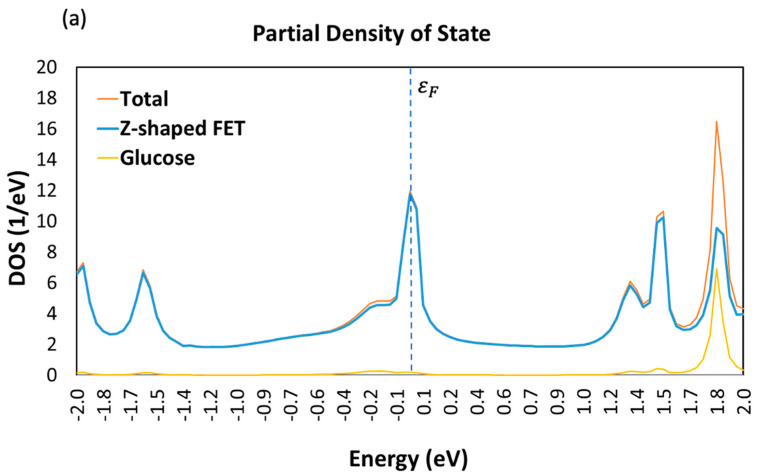
Total and partial density of states (DOS) of GNR-FET in the presence of (**a**) Glucose molecule; (**b**) Fructose molecule; and (**c**) Xylose molecule.

**Figure 7 sensors-23-03010-f007:**
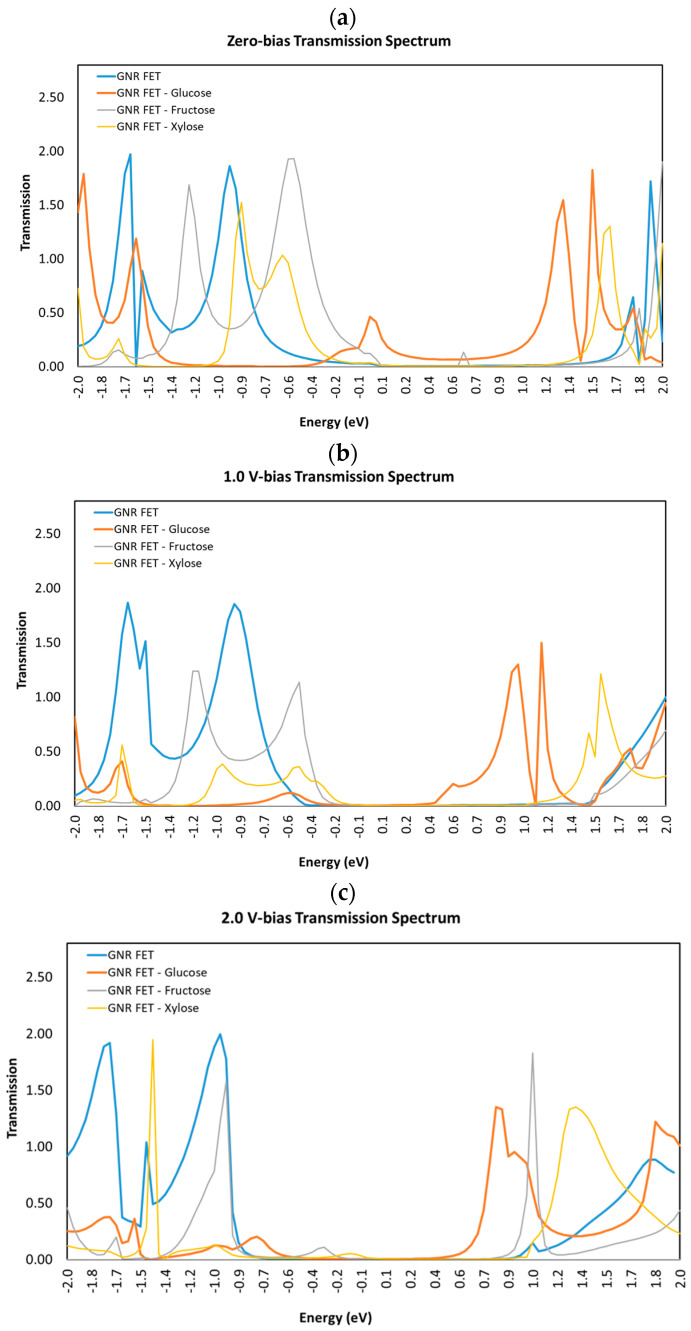
Transmission spectra T(E) for GNR-FET sensor without and with each of the three different sugar molecules (**a**) Bias voltage = 0 V, (**b**) Bias voltage = 1.0 V, and (**c**) Bias voltage = 2.0 V.

**Figure 8 sensors-23-03010-f008:**
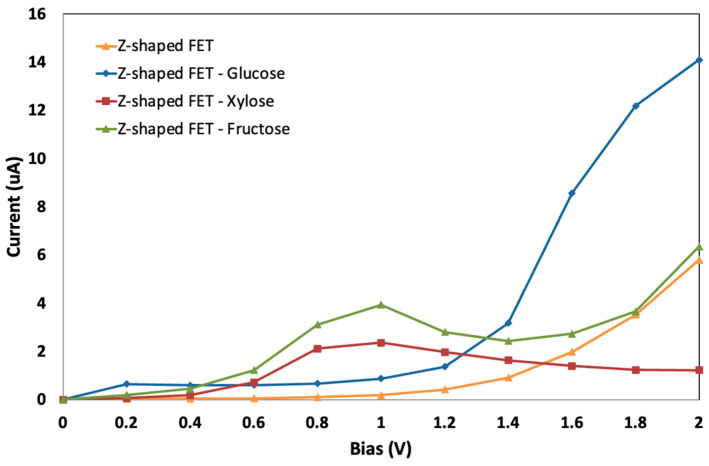
Current-Voltage characteristics for the GNR FET (orange), for the GNR FET with glucose (blue), for the GNR FET with xylose (red) and for the GNR FET with fructose (green).

**Figure 9 sensors-23-03010-f009:**
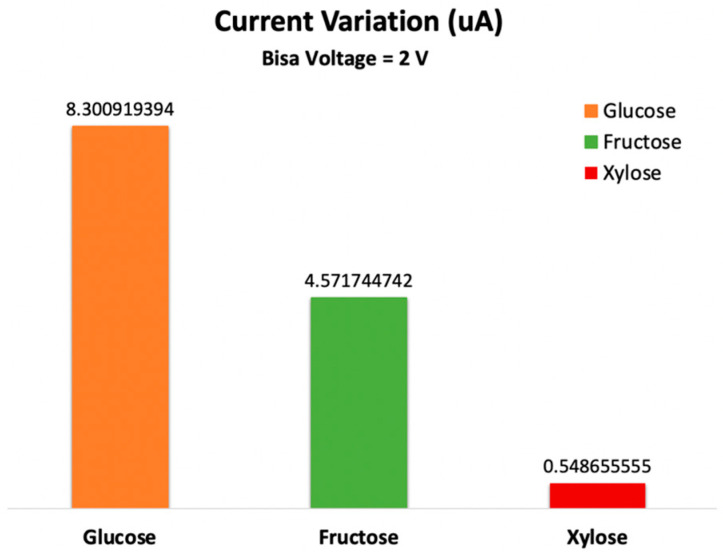
Change in electrical drain current for the GNR-FET due to the different types of the sugar molecules.

**Figure 10 sensors-23-03010-f010:**
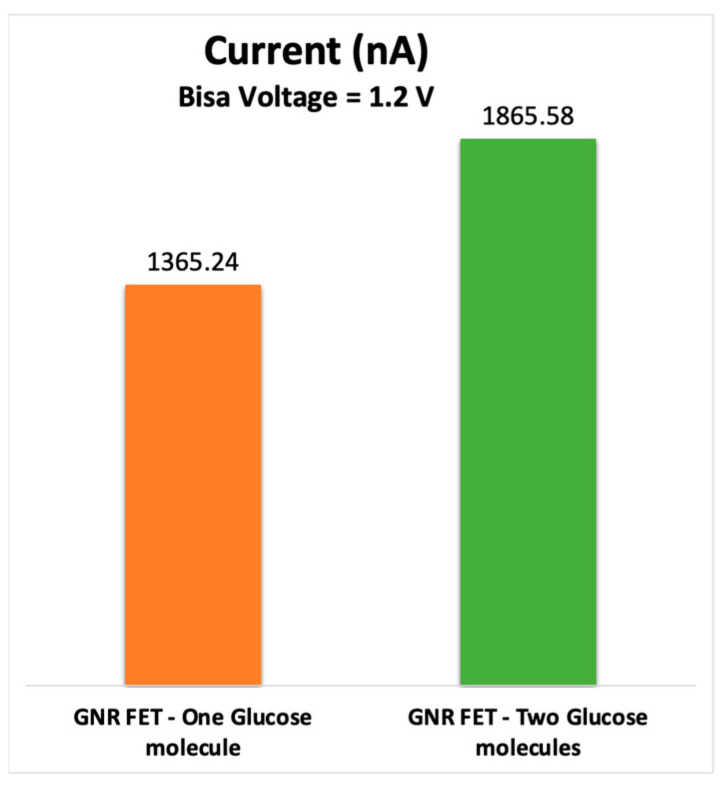
Electrical drain current for the GNR-FET due to one glucose molecule and two glucose molecules.

## Data Availability

All the data generated or analyzed during this study are included in this published article.

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
