# Peer review of "Graphene Nanoribbon Field Effect Transistor Simulations for the Detection of Sugar Molecules: Semi-Empirical Modeling"

_sensors, 2023, doi:10.3390/s23063010_

Round 1
Reviewer 1 Report
The manuscript entitled “Graphene Nanoribbon Field Effect Transistor Simulations for the Detection of Sugar Molecules: Semi-Empirical Modeling” reports an interesting study of the transport properties of sugar molecules adsorbed in a graphene nanoribbon field effect transistor. Those systems are designed to form glucose sensors.
From my point of view, the manuscript deserves to be published on sensors. However, many issues must be solved, specifically, and the discussion and conclusions must be improved. For example,
1. In the Density of States analysis, there is a lot of missing information. Where is the Fermi energy? What happens in the virtual zone? Do the molecules affect the DOS? A deeper analysis is needed
2. Much information and analysis are missing about the transmission spectrum. Why is the transmission not symmetrical? Is there a connection between the transmission spectrum and DOS?
3. What is the connection between the transmission spectrum and the current-voltage plot?
4. In lines 249-251, the authors mentioned that the sugar molecules act as charge carriers, specifically donors, and the concentration of the graphene charge carriers increases. This fact is very interesting. It is important to prove this fact. In what way do they conclude that molecules act as donors? What happens with the Fermi level in that case? Is there a type of p-doping or n-doping?
Author Response
Manuscript title: Graphene Nanoribbon Field Effect Transistor Simulations for the
Detection of Sugar Molecules: Semi-Empirical Modeling.
Manuscript number: Sensors-2198910
Dear Mr. Harrison Li,
We are grateful to the reviewers for the valuable comments, which helped enhance the quality of our manuscript. Our responses to the reviewer's comments are shown in the attached document in blue text and the changes are highlighted in the manuscript.
Best regards,
Falah
Response to the comments:
- Reviewer 1:
The manuscript entitled “Graphene Nanoribbon Field Effect Transistor Simulations for the Detection of Sugar Molecules: Semi-Empirical Modeling” reports an interesting study of the transport properties of sugar molecules adsorbed in a graphene nanoribbon field effect transistor. Those systems are designed to form glucose sensors.
From my point of view, the manuscript deserves to be published on sensors. However, many issues must be solved, specifically, and the discussion and conclusions must be improved. For example,
Response: We appreciate the Referee's positive feedback on our manuscript. We have taken the reviewer's insights and guidance into consideration and have made significant improvements to the new version of the manuscript.

Reviewer 2 Report
In this work, the authors present simulated results on graphene nanoribbon FET (GNR-FET) for sensing of Sugar molecules, namely, glucose, fructose and xylose. However, the reviewer has the following comments/queries on the manuscript.
(1) In the paragraph from line 83 to 88, authors mention several mechanisms of graphene FET that causes the current to change. However, the authors does not mention specifically which of these mechanisms play role in the change of current in the presence of the sugar molecules. Is this doping effect by the sugar molecules of change of conductance? Please elaborate with respect to the chemical structure of those three sugar molecules.
(2) In the introduction section, the authors should elaborate more on the novelty of this work compared to current state of the art and elaborate how this study will be important to the research community.
(3) A common graphene FET characteristic is the V-shaped Id-Vgs transfer characteristics with minimum current at the Dirac point. Are the authors be able to generate that characteristics curve for the nanoribbon GFET? If yes, can the shift of the Dirac point be used as a sensing method?
(4) Are the authors able to present any numerical values for any figure of merits for the GNR FET, i.e. transconductance or on-off ratio, etc.?
(5) Also, the authors claim the change in current in Figure 7 is due to the presence of the sugar molecules. However, to confirm this claim, the authors must present more data of concentration dependent change of current to show that this current change is in fact specific to the presence of these sugar molecules.
(6) Moreover, the authors should calculate the limit of detection/sensitivity and compare with other similar work for comparison.
(7) Figure 8 shows that the sensor is highly sensitive for glucose. The authors should explain why it is more sensitive to glucose compared to the other 2 sugar molecules.
(8) Also, in the conclusion, the authors claim that the sensor has the potential to accurately detect sugar molecules. However, the authors do not have any results that corroborate this claim. The results presented so far just presents a proof-of-concept study of using GNR-FET for sensing sugar molecules, but none of the results tells give any idea on the accuracy.
Author Response
Manuscript title: Graphene Nanoribbon Field Effect Transistor Simulations for the
Detection of Sugar Molecules: Semi-Empirical Modeling.
Manuscript number: Sensors-2198910
Dear Mr. Harrison Li,
We are grateful to the reviewers for the valuable comments, which helped enhance the quality of our manuscript. Our responses to the reviewer's comments are shown in the attached document in blue text and the changes are highlighted in the manuscript.
Best regards,
Falah
Response to the comments:
- Reviewer 2:
In this work, the authors present simulated results on graphene nanoribbon FET (GNR-FET) for sensing of Sugar molecules, namely, glucose, fructose and xylose. However, the reviewer has the following comments/queries on the manuscript.
Response: We appreciate the feedback provided by the referee on our manuscript. We have taken their recommendations into consideration and have made improvements to the updated version of the manuscript.

Round 2
Reviewer 1 Report
I appreciate all the changes and explanations about the issues raised previously. With the improvement, the manuscript is ready to be published.
Reviewer 2 Report
The reviewers have answered all the review queries. The manuscript can be accepted.